# Colistin Monotherapy versus Colistin plus Sitafloxacin for Therapy of Carbapenem-Resistant *Acinetobacter baumannii* Infections: A Preliminary Study

**DOI:** 10.3390/antibiotics11121707

**Published:** 2022-11-26

**Authors:** Rujipas Sirijatuphat, Supawas Thawornkaew, Darat Ruangkriengsin, Visanu Thamlikitkul

**Affiliations:** Department of Medicine, Faculty of Medicine Siriraj Hospital, Mahidol University, Bangkok 10700, Thailand

**Keywords:** colistin, sitafloxacin, carbapenem-resistant, *Acinetobacter baumannii*

## Abstract

The in vitro study of sitafloxacin against carbapenem-resistant (CR) *Acinetobacter baumannii* demonstrated activity against most strains of CR *A. baumannii*, and the combination of colistin and sitafloxacin showed an in vitro synergistic effect against CR *A. baumannii*. This study aimed to compare efficacy and safety between colistin plus sitafloxacin with colistin alone for therapy for CR *A. baumannii* infection. This randomized controlled trial enrolled 56 patients with CR *A. baumannii* infection (28/group) during 2018–2021, and the treatment duration was 7–14 days. The study outcomes were 28-day mortality, clinical and microbiological responses, and adverse events. There was no significant difference in 28-day mortality between groups (32.1% combination vs. 32.1% monotherapy, *p* = 1.000). Favorable clinical response at the end of treatment was comparable between groups (81.5% combination vs. 77.8% monotherapy, *p* = 0.788). Microbiological response at the end of treatment was also comparable between groups (73.1% combination vs. 74.1% monotherapy, *p* = 0.934). Acute kidney injury was found in 53.8% of the combination group, and in 45.8% of the monotherapy group (*p* = 0.571). In conclusion, there was no significant difference in 28-day mortality between the colistin monotherapy and the colistin plus sitafloxacin groups. There was also no significant difference in adverse events between groups.

## 1. Introduction

Carbapenem-resistant (CR) *Acinetobacter baumannii* is a dangerous pathogen that causes harmful nosocomial infections worldwide, including in Thailand. The prevalence of CR *A. baumannii* is continuously increasing, and the mortality of infections caused by CR *A. baumannii* is still high. In Thailand, the National Antimicrobial Resistance Surveillance Center, Thailand (NARST) collected the information on antibiotic susceptibility of common bacteria causing infections in Thai patients in 2020 and reported that approximately 80% of *A. baumannii* isolates from the patients in the intensive care unit (ICU) were resistant to carbapenems [1]. Although there are some antibacterial drugs (such as polymyxins) that are commonly used to treat CR *A. baumannii* infections in low- and middle-income countries, most of them lack sufficient effectiveness for treating CR *A. baumannii* infections. CR *A. baumannii* has been classified as a priority antibiotic-resistant pathogen by the World Health Organization (WHO), for which the development of new antibiotics is urgently needed [2]. 

Colistin (polymyxin E) was discovered in the 1950s and then abandoned in the 1980s due to its toxicity and the availability of many safer and more effective antibiotics against Gram-negative bacterial infections, including cephalosporins and carbapenems. However, colistin was readopted as a therapy for CR *A. baumannii* infections due to the lack of effective antibiotics to treat CR *A. baumannii* infections [3]. Colistin is now a first-line antibiotic for therapy of infections caused by CR Gram-negative bacteria, including *A. baumannii*. In Thailand, colistin-resistant *A. baumannii* isolates were rarely observed. Colistin was reported to decrease the mortality of patients with CR *A. baumannii* infection by nearly half when compared with the therapy with antibiotics that were inactive against CR *A. baumannii*, but the mortality rate in patients being treated with this drug was still as high as 46% [4,5]. Nephrotoxicity is the major adverse effect reported with the use of colistin for the therapy of infections caused by CR Gram-negative bacteria. Many studies reported the prevalence of colistin-associated nephrotoxicity in 20–69% of the patients who received colistin, and the incidence of colistin-induces nephrotoxicity was dose-dependent, and higher daily doses and the accumulative amount of colistin were associated with a higher rate of nephrotoxicity [6]. 

Many antibiotic combination therapies have been studied for the treatment of patients with CR *A. baumannii* infections, but those studies revealed no significant differences in mortality compared with monotherapy. By way of example, among patients with severe CR Gram-negative bacilli infection (77% of study patients had CR *A. baumannii* infection), colistin plus meropenem and colistin monotherapy showed comparable clinical failure and 28-day mortality [7]. The combination of rifampicin or fosfomycin with colistin showed a favorable microbiological response, but the mortality rates of combination therapy patients were not significantly different from the rates observed in colistin monotherapy patients [8,9].

Fluoroquinolones are common antibiotics that have broad spectrum activity against both Gram-negative and Gram-positive pathogens. Sitafloxacin is a newly developed fluoroquinolone that has potent activity against a wide spectrum of bacteria by inhibiting DNA gyrase and topoisomerase IV [10,11]. This antimicrobial agent shows in vitro activity against a broad range of activity against Gram-positive and Gram-negative bacteria, as well as anaerobic bacteria and atypical pathogens [12]. Sitafloxacin is approved in Japan for the treatment of bacterial infections (i.e., acute otitis media, sinusitis, periodontitis, pericoronitis, osteitis of the jaw, laryngopharyngitis, tonsillitis, acute bronchitis, pneumonia, secondary infections of chronic respiratory disease, cystitis, acute pyelonephritis, urethritis, and cervicitis) caused by sitafloxacin-susceptible strains of *Staphylococcus* spp., *Streptococcus* spp., *S. pneumoniae*, *Enterococcus* spp., *Moraxella catarrhalis*, *Escherichia coli*, *Klebsiella* spp., *Citrobacter* spp., *Enterobacter* spp., *Serratia* spp., *Proteus* spp., *Morganella morganii*, *Haemophilus influenzae*, *Pseudomonas aeruginosa*, *Peptostreptococcus* spp., *Prevotella* spp., *Porphyromonas* spp., *Fusobacterium* spp., *Legionella pneumophila*, *Chlamydia trachomatis*, *Chlamydophila pneumoniae*, and *Mycoplasma pneumoniae* [12]. Sitafloxacin is also approved in Thailand for the treatment of respiratory tract infections and urinary tract infections [13].

Sitafloxacin is only available as 50 mg tablets for oral administration. The pharmacokinetic profile of sitafloxacin was favorable [12,13]. Oral administration of 100 mg of sitafloxacin was rapidly absorbed with high drug bioavailability. Food intake did not affect the proportion and rate of absorption. The mean maximum concentration of 100 mg oral sitafloxacin in serum was approximately 1 mg/L, with an elimination half-life of 5.7 h. Sitafloxacin was primarily eliminated by the kidney. The area under the concentration–time curve was 5.5 mg·h/L. Serum-protein-binding of the drug was approximately 50%. The apparent volume of distribution was 2.5 L/kg, and other previous studies also revealed that sitafloxacin exhibited good distribution into various tissues. The recommended dosages of oral sitafloxacin in adults is 50 mg twice daily. If patients are suspected of having a poor clinical response to oral sitafloxacin 50 mg twice daily, the dosage of oral sitafloxacin may be increased to 100 mg twice daily. The dosage adjustment of sitafloxacin is recommended in patients with moderate or severe renal impairment [12]. Sitafloxacin administration was safe and well-tolerated. Diarrhea and the abnormal liver function test are the most commonly reported adverse events in patients receiving oral sitafloxacin and are usually mild [12].

Sitafloxacin has been shown to have good in vitro activity against many nosocomial pathogens, especially *A. baumannii* [13,14,15,16]. For example, in a previous study that was conducted in Taiwan including 167 blood isolates of CR *Acinetobacter* spp., sitafloxacin had a significantly lower minimum inhibitory concentration (MIC) compared with ciprofloxacin and levofloxacin, and the rate of resistance to sitafloxacin was significantly lower than that to ciprofloxacin, levofloxacin, and tigecycline [15]. In a recent study in China including 50 strains of extensively-drug resistant (XDR) *A. baumannii* isolated from clinical specimens, the susceptibility rate of sitafloxacin was 92% originally, and the susceptibility rate increased to 96% for the combination of sitafloxacin and sulbactam [17]. In a recent study in Thailand in 2016 that included clinical isolates of bacteria from patients with urinary tract infections and lower respiratory tract infections, sitafloxacin was more active than ceftazidime, piperacillin-tazobactam, imipenem, and meropenem against *A. baumannii* isolates, including *A. baumannii* isolates that were resistant to carbapenems. The activity of sitafloxacin against *A. baumannii* isolates was comparable to that of tigecycline, but it was less active than colistin [16].

A previous study that was conducted at Siriraj Hospital (Bangkok, Thailand) showed a correlation between the minimum inhibitory concentrations (MICs) and the inhibition zone diameters of sitafloxacin against CR Gram-negative bacilli, including CR *A. baumannii* [18]. In that study, the inhibition zone diameters ≥16 mm and ≥18 mm were proposed to be appropriate breakpoints for indicating the susceptibility of resistant Gram-negative bacilli isolated from urine and blood, respectively. Using a cut-off minimal inhibitory concentration (MIC) of sitafloxacin ≤2 mg/L, the rate of susceptibility of CR *A. baumannii* to sitafloxacin was 91.4% [13]. The combination of colistin and sitafloxacin demonstrated an in vitro synergistic effect against *A. baumannii* in previous studies [19,20]. 

Therefore, sitafloxacin is an interesting antibiotic option for the treatment of CR *A. baumannii* infections.

Since there is no proven effective colistin combination therapy for the treatment of patients with CR *A. baumannii* infection, a combination of sitafloxacin and colistin might improve clinical and microbiological outcomes. The aim of this study was to compare 28-day all-cause mortality, clinical and microbiological responses, and adverse events of colistin plus sitafloxacin versus colistin alone for the treatment of CR *A. baumannii* infections.

## 2. Materials and Methods

This preliminary open-label randomized controlled study was conducted at Siriraj Hospital, which is a 2300-bed tertiary care university hospital that is located in Bangkok, Thailand, during December 2018 to December 2021. The protocol for this study was approved by the Siriraj Institutional Review Board (SIRB) of the Faculty of Medicine Siriraj Hospital, Mahidol University, Bangkok, Thailand, and all patients or their legal representatives provided written informed consent to participate in this study. All aspects of this study were in compliance with the principles and guidelines set forth in the 1964 Declaration of Helsinki and all of its subsequent amendments. Eligible participants were hospitalized patients aged ≥18 years with laboratory confirmed CR *A. baumannii* infections who were able to take/tolerate enteral nutrition (oral or tube feeding), and who might need to commence colistin within 48 h of enrollment. Pregnant and breastfeeding women, patients with major contraindication to colistin or fluroquinolones (drug allergy), patients with a suspected or confirmed case of tuberculosis infection, and patients with ongoing liver impairment (≥3 times the upper limit of AST and/or ALT) were excluded. At Siriraj Hospital (Bangkok, Thailand), the identification of *A. baumannii* isolate and the determination of its antimicrobial susceptibility profiles were based on Clinical and Laboratory Standards Institute (CLSI) recommendations. The colistin susceptibility test using the broth microdilution method was performed in all *A. baumannii* isolates, and patients with colistin-resistant *A. baumannii* infections were not included in this study.

Eligible patients were randomized by blocks of 4-randomization to receive a combination of intravenous colistin at a loading dose of 300 mg followed by a maintenance dose (monotherapy group), or colistin at the aforementioned dose plus oral sitafloxacin at a dosage of 200 mg/d (combination group). The maintenance doses of colistin and sitafloxacin were adjusted according to the renal function (estimated creatinine clearance (CrCl) by the Cockcroft–Gault equation) of each patient. In Thailand, the maintenance dose of colistin is suggested as shown [21]. Patients with CrCl >50 mL/min will be given 300 mg/d of colistin, in 2–3 divided doses. Patients with CrCl 41–50 mL/min will be given 225–300 mg/d of colistin, in 2–3 divided doses. Patients with CrCl 31–40 mL/min will be given 150–200 mg/d of colistin, in 2 divided doses. Patients with CrCl 21–30 mL/min will be given 150 mg/d of colistin, in one or two divided doses. When CrCl ≤20 mL/min, the patient will receive 100 mg/d of colistin every 24 h. The high dose of sitafloxacin should be required for the treatment of CR *A. baumannii* infections; therefore, 200 mg/d oral sitafloxacin is recommended in this study because it is the maximum recommended daily dosage of sitafloxacin in Thailand and Japan [12,13]. For the renal dosage adjustment of sitafloxacin [12], patients with CrCl >50 mL/min will be given 200 mg/d of sitafloxacin, in 2 divided doses. Patients with CrCl 30–50 mL/min will be given 100 mg/d of sitafloxacin, in one dose. When CrCl <30 mL/min, the patient will receive 100 mg of sitafloxacin every 48 h. The duration of treatment was 7 to 14 days depending on the site of infection and overall patient response. Daily clinical assessment was closely observed from enrollment until the administration of antibiotics was discontinued or the patients died or the patient was discharged from the hospital. The microbiological culture of the specimen taken from infection site was taken on day 3 after initiating the study treatment, and then again at the end of treatment. Clinical outcomes were assessed at day 3 after initiation of the study treatment, at the end of study treatment, and 28 days after treatment with the study drug(s). The clinical outcome was classified as a favorable outcome (cure or improvement) or unfavorable outcome (progression of infection or death). The microbiological response was classified as the eradication (absence CR *A. baumannii* from repeat culture of the sample collected from the infection site) or persistance of CR *A. baumannii* at the infection site. Liver and renal function tests were performed at least twice a week. Acute kidney injury was defined according to risk (decline of glomerular filtration rate of 25%, increased serum creatinine 1.5 times, or urine production of <0.5 mL/kg/h for 6 h), injury (decline of glomerular filtration rate of 50%, a doubling of serum creatinine, or urine production of <0.5 mL/kg/h for 12 h), failure (decline of glomerular filtration rate of 75%, increased serum creatinine 3 times, serum creatinine ≥4 mg/dL (with acute rise of >0.5 mg/dL), urine production of <0.3 mL/kg/h for 24 h, or anuria for 12 h), loss of kidney function (complete loss of kidney function >4 weeks), and end-stage kidney disease (complete loss of kidney function >3 months) (RIFLE) classification. Acute kidney injury was determined as at least the injury category of the RIFLE classification. Acute hepatitis was defined as an increase of 5 times the upper limit of aspartate transaminase (AST) and/or alanine transaminase (ALT), or symptomatic patients with 3 times the upper limit of AST and/or ALT.

### Sample Size Calculation and Statistical Analysis

The 28-day mortality rate of patients with CR *A. baumannii* infection treated with colistin alone was 46%, and the 28-day survival rate was 54% [3]. Using this data, assuming that combination therapy will increase survival by 50% (from 54% to 81%) when compared with colistin monotherapy, and allowing for a 5% type I error and a 20% type II error, our sample size formula calculated a minimum of 47 patients per group.

The tests that were used to compare the data were the chi-squared test or Fisher’s exact test for categorical data, and Student’s t-test (for normally distributed data) or the Mann–Whitney U test (for non-normally distributed data) for continuous data. The results of those comparisons are reported as the number and percentage for categorical data, and as mean ± standard deviation (SD) and median and interquartile range (IQR) for normally distributed and non-normally distributed continuous data, respectively. Univariate and multivariate analyses were performed to identify risk factors significantly and independently associated with 28-day mortality. The results of those analyses are given as the odds ratio (OR) and 95% confidence interval (CI) and adjusted OR and 95% CI, respectively. A *p*-value of ≤0.05 was considered statistically significant for all tests. All statistical analyses were performed using the SPSS for Windows software package, version 18 (SPSS, Inc., Chicago, IL, USA).

## 3. Results

Fifty-six patients were enrolled in this study (approximately 60% of the minimum sample size determined by our sample size calculation), with 28 patients randomized to the combination group and the other 28 patients to the monotherapy group. The baseline characteristics of patients were similar between groups, except for more chronic kidney disease (CKD) and mechanical ventilation in the colistin monotherapy group (Table 1).

Most patients were male and of advanced age. Both groups had multiple underlying diseases, a high rate of ICU admission, and similar Acute Physiology and Chronic Health Evaluation II (APACHE II) scores. The colistin monotherapy group had significantly more CKD patients compared to the combination group (32.1% and 7.1%, *p* = 0.019), and significantly more patients receiving mechanical ventilation compared to the combination group (92.9% and 71.4%, *p* = 0.036). Most patients in both groups (71% in each group) had no concurrent coinfection(s) while receiving study treatment. Most patients had pneumonia, both among overall patients and in each of the two study groups. Other common sites of infection included blood stream infection (BSI), urinary tract infection (UTI), intra-abdominal infection (IAI), and soft tissue and skin infection (SSI). Among the less prevalent sites of infection, three patients had acute tracheobronchitis, one patient had otitis media, and one patient had empyema thoracis. The dose and duration of colistin and sitafloxacin are shown in Table 2. The average dose ± standard deviation of colistin given to all patients was 4.2 ± 1.9 mg/kg/day, and the average dose of colistin was not significantly different between groups. The duration of antibiotic treatment for CR *A. baumannii* infections was comparable between both groups.

The 28-day mortality rate was 32.1% in both treatment groups (*p* = 1.000). The favorable clinical outcome rate among all study patients was 55.4% and 79.6% at 72 h after the start of study treatment and at the end of treatment, respectively. There was no significant difference in the rate of favorable clinical outcome between the combination and monotherapy groups at either the 72 h after the start of study treatment time point (57.1% vs. 53.6%, respectively; *p* = 0.788) or at the end of treatment (81.5% vs. 77.8%, respectively; *p* = 0.735) (Table 3). The microbiological response in the combination and monotherapy groups was comparable at 72 h after the start of treatment (44.4% vs. 57.1%, respectively; *p* = 0.346) and at the end of treatment (73.1% vs. 74.1%, respectively; *p* = 0.934). The median length of hospital stay in all patients with CR *A. baumannii* infections was 46 days, and the median length of hospital stay was not significantly different between both groups.

Acute kidney injury was found in 53.8% of patients in the combination group, and in 45.8% of patients in the colistin monotherapy group (*p* = 0.571). Hepatitis and rash were observed in one patient who received combination therapy.

The results of the univariate and multivariate analysis to identify factors significantly and independently associated with 28-day mortality are shown in Table 4. The factors independently associated with 28-day mortality were Sequential Organ Failure Assessment (SOFA) score (adjusted odds ratio (aOR): 1.62, 95% confidence interval (CI): 1.14–2.30; *p* = 0.007), and cerebrovascular disease (aOR: 28.20, 95% CI: 3.44–230.76; *p* = 0.002).

## 4. Discussion

The combinations of different antibiotics (i.e., carbapenems, rifampicin, fosfomycin, ampicillin-sulbatam, and vancomycin) with colistin have been proposed and studied for treating the patients with CR *A. baumannii* infections. However, a reduction in mortality when using colistin combined with other agents was not clearly observed when compared with colistin alone [7,8,9,22,23,24]. The present study selected sitafloxacin to combine with colistin because sitafloxacin was found to be active against most isolates of CR *A. baumannii*, and colistin combined with sitafloxacin demonstrated an in vitro synergistic effect against CR *A. baumannii* [19,20]. However, the results of our randomized controlled study in 56 patients with CR *A. baumannii* infection showed the combination therapy not to be superior to monotherapy relative to 28-day mortality. There are some reasons that may help to explain this finding. First, we were only able to enroll approximately 60% of the calculated minimum sample size per group, and our sample size calculation assumed a 50% reduction in mortality in the combination group, which was likely too high. We decided to stop this study after the enrollment of 60% of the patients because the mortality rates of the patients in both groups were identical and it was extremely unlikely to observe a 50% reduction in the mortality rate of the patients in the combination therapy group. Second, although most patients’ characteristics were similar between groups, the colistin monotherapy group had significantly more CKD and mechanical ventilation, which suggests a more serious condition of patients in the monotherapy group than in the combination therapy group. Third, the dose and route of administration of sitafloxacin might both be inappropriate. The parenteral form of sitafloxacin was not available, so the oral form of this drug had to be used instead. The bioavailability of oral sitafloxacin was reported to be approximately 89% [25], and the maximum concentration of sitafloxacin at 100 mg/day (mean ± standard deviation) was only 1.17 ± 0.45 mg/L [26]. Therefore, the concentration of sitafloxacin may have been too low, especially the concentration of sitafloxacin in the lungs. Therefore, an appropriate dose of parenteral sitafloxacin might be more effective than the 200 mg/d oral sitafloxacin that was used in this study. Further study comparing these two drug regimens, except using parenteral sitafloxacin instead of oral sitafloxacin, in a sufficiently large study population of patients with CR *A. baumannii* infection is needed. Fourth and last, our study findings are consistent with the results of other clinical studies. Although in vitro studies showed that the combination of various antibiotics with colistin conferred a synergistic effect against *A. baumannii* isolates, the favorable clinical outcomes could not be obtained by using these combination treatments in patients with CR *A. baumannii* infections in the clinical studies [7,8,9,22,23,24]. It is important to note that patients with CR *A. baumannii* infections generally have severely acute medical problems and several underlying illnesses. These factors have a significant effect on clinical outcomes, including clinical improvement and all-cause mortality. Moreover, since patients with CR *A. baumannii* infections frequently have other concurrent infections (i.e., bacteria or fungi), the relative contributions of CR *A.baumanni* infections versus other infections on the clinical outcomes and mortality are difficult to determine [27].

This study has some mentionable limitations. First, our data were collected from a single center, which is a university-based national tertiary referral center that is routinely referred complex cases that are thought to be untreatable at less sophisticated levels of care. This suggests that some aspects of our data may not be immediately generalizable to other medical care settings. Second, we were not able to satisfy the minimum number of patients per group as prescribed by our sample size calculation. We enrolled approximately 60% of the prescribed sample size in each of the two study groups. As such, it is very possible that our study lacked the statistical power to identify all significant differences and associations between groups in this study. Third and last, this study was conducted in Thailand, and the patient characteristics and distribution of resistance mechanisms of CR *A. baumannii* may differ according to local epidemiology and the patient characteristics, therefore, it should be acknowledged that this factor could influence the effect of combination therapy. 

According to our study results, the role of the combination of colistin and sitafloxacin in the treatment of CR *A. baumannii* infections is not established. This is important information for physicians, pharmacists, and healthcare professionals because unnecessary combination therapies can increase the possibility of side effects, *Clostridioides difficile* infection, cost of antibiotic treatment, and emergence of antimicrobial resistant organisms [27,28]. 

## 5. Conclusions

The results of this study revealed no significant difference in 28-day mortality between the colistin monotherapy and the colistin plus sitafloxacin groups for the therapy of CR *A. baumannii* infections in Thai patients. There was also no significant difference in adverse events between groups. The factors found to be independently associated with 28-day mortality were SOFA score and cerebrovascular disease.

## Figures and Tables

**Table 1 antibiotics-11-01707-t001:** The baseline characteristics of all study patients and compared between the combination group and the monotherapy group.

Characteristics	Total	Colistin-Sitafloxacin	Colistin Monotherapy	*p*-Value
	(*n* = 56)	(*n* = 28)	(*n* = 28)	
Female gender, *n* (%)	22 (39.3%)	10 (35.7%)	12 (42.9%)	0.584
Age (years); mean ± SD	69.2 ± 12.2	69.3 ± 13.2	69.1 ± 11.4	0.957
Comorbidities, *n* (%)				
- DM	18 (32.1%)	8 (28.6%)	10 (35.7%)	0.567
- HT	29 (51.8%)	17 (60.7%)	12 (42.9%)	0.181
- CKD	11 (19.6%)	2 (7.1%)	9 (32.1%)	0.019
- CAD	9 (16.1%)	4 (14.3%)	5 (17.9%)	1
- CVD	11 (19.6%)	5 (17.9%)	6 (21.4%)	0.737
- COPD	5 (8.9%)	3 (10.7%)	2 (7.1%)	1
- Cancer	21 (37.5%)	12 (42.9%)	9 (32.1%)	0.408
APACHE II score, mean ± SD	18.1 ± 5.2	18.1 ± 5.6	18.0 ± 4.9	0.94
SOFA score, median (IQR)	5.0 (4.0, 7.0)	6.5 (4.0, 7.8)	5.0 (3.3, 6.8)	0.39
Mechanical ventilation, *n* (%)	46 (82.1%)	20 (71.4%)	26 (92.9%)	0.036
ICU admission, *n* (%)	29 (51.8%)	12 (42.9%)	17 (60.7%)	0.181
Type of infection, *n* (%)				
- Pneumonia	33 (58.9%)	13 (46.4%)	20 (71.4%)	0.051
- Bloodstream infection	12 (21.4%)	6 (21.4%)	6 (21.4%)	1
- Urinary tract infection	3 (5.4%)	1 (3.6%)	2 (7.1%)	0.553
- Intra-abdominal infection	2 (3.6%)	2 (7.1%)	0 (0.0%)	0.15
- Soft tissue/skin infection	1 (1.8%)	1 (3.6%)	0 (0.0%)	0.313
- Other ^a^	5 (8.9%)	5 (17.9%)	0 (0.0%)	0.019
Coinfection, *n* (%)	16 (28.6%)	8 (28.6%)	8 (28.6%)	1
- *P. aeruginosa*	7 (12.5%)	2 (7.1%)	5 (17.9%)	0.422
- *S. maltophilia*	6 (10.7%)	3 (10.7%)	3 (10.7%)	1
- Other ^b^	7 (12.5%)	5 (17.9%)	2 (7.1%)	0.422
No coinfection; *n* (%)	40 (71.4%)	20 (71.4%)	20 (71.4%)	1
Concurrent antibiotics, *n* (%)	20 (35.7%)	8 (28.6%)	12 (42.9%)	0.265
- Carbapenems	6 (10.7%)	2 (7.1%)	4 (14.3%)	0.669
- Piperacillin/tazobactam	5 (8.9%)	0 (0.0%)	5 (17.9%)	0.051
- Vancomycin	6 (10.7%)	3 (10.7%)	3 (10.7%)	1
- Aminoglycosides	1 (1.8%)	1 (3.6%)	0 (0.0%)	0.313
- Other ^c^	6 (10.7%)	5 (17.9%)	1 (3.6%)	0.193

A *p*-value < 0.05 indicates statistical significance; ^a^ other site of infections, acute tracheobronchitis, otitis media, and empyema thoracis; ^b^ other coinfections, *Enterococcus cloacae, Enterococcus faecium,* vancomycin-resistant enterococci, *Pneumocystis jerovecii,* cytomegalovirus, mixed bacteria, non-fermenter Gram-negative bacteria; ^c^ other concurrent antibiotics, tigecycline, levofloxacin, co-trimoxazole, ceftriaxone. Abbreviations: SD, standard deviation; IQR, interquartile range; DM, diabetes mellitus; HT, hypertension; CKD, chronic kidney disease; CAD, coronary artery disease; CVD, cerebrovascular disease; COPD, chronic obstructive pulmonary disease; *P., Pseudomonas; S., Stenotrophomonas;* SOFA, Sequential Organ Failure Assessment; APACHE, Acute Physiology and Chronic Health Evaluation.

**Table 2 antibiotics-11-01707-t002:** Dose and duration of colistin and sitafloxacin among total patients, and compared between the combination and monotherapy groups.

Parameters	Total (*n* = 56)	Colistin-Sitafloxacin (*n* = 28)	Colistin Monotherapy (*n* = 28)	*p*-Value
Dose of colistin (mg/kg/d), mean ± SD	4.2 ± 1.9	4.0 ± 1.7	4.3 ± 2.1	0.55
Duration of colistin (days), median (IQR)	10.0 (7.0, 12.8)	9.5 (7.0,14.0)	10.0 (7.0, 12.0)	0.987
Dose of sitafloxacin (mg/d), mean ± SD	160.7 ± 49.7	160.7 ± 49.7	-	-
Duration of sitafloxacin (days), median (IQR)	7.0 (5.0, 7.0)	7.0 (5.0, 7.0)	-	-

A *p*-value < 0.05 indicates statistical significance. Abbreviations: SD, standard deviation; IQR, interquartile range.

**Table 3 antibiotics-11-01707-t003:** Clinical outcomes, microbiological response, and adverse events among total patients, and compared between the combination and monotherapy groups.

Parameters	Total (*n* = 56)	Colistin-Sitafloxacin (*n* = 28)	Colistin Monotherapy (*n* = 28)	*p*-Value
**Clinical response, *n* (%)**				
First 72 h				
- No response	25 (44.6%)	12 (42.9%)	13 (46.4%)	0.788
- Favorable response	31 (55.4%)	16 (57.1%)	15 (53.6%)	
End of treatment				
- No response	11 (20.4%)	5 (18.5%)	6 (22.2%)	0.735
- Favorable response	43 (79.6%)	22 (81.5%)	21 (77.8%)	
**Microbiological response, *n* (%)**				
First 72 h				
- Persistence	27 (49.1%)	15 (55.6%)	12 (42.9%)	0.346
- Eradication	28 (50.9%)	12 (44.4%)	16 (57.1%)	
End of treatment				
- Persistence	14 (26.4%)	7 (26.9%)	7 (25.9%)	0.934
- Eradication	39 (73.6%)	19 (73.1%)	20 (74.1%)	
Adverse events, *n* (%)				
- AKI (RIFLE) ^d^	25 (50.0%)	14 (53.8%)	11 (45.8%)	0.571
- Hepatitis	1 (3.6%)	1 (3.6%)	0 (0.0%)	-
- Rash	1 (3.6%)	1 (3.6%)	0 (0.0%)	-
28-day mortality, *n* (%)	18 (32.1%)	9 (32.1%)	9 (32.1%)	1
Length of hospital stay, median (IQR)	46.0 (33.0, 70.0)	44.5 (32.3, 71.5)	48.0 (34.3, 65.3)	0.787

A *p*-value < 0.05 indicates statistical significance; ^d^ RIFLE, risk, injury, failure, loss of kidney function, and end-stage kidney disease. Abbreviation: IQR, interquartile range.

**Table 4 antibiotics-11-01707-t004:** Univariate and multivariate analysis for factors independently associated with 28-day mortality.

Factors	Univariate Analysis		Multivariate Analysis	
	OR	*p*-Value	Adjusted OR	*p*-Value
	(95% CI)		(95% CI)	
Female gender	1.37	0.587	1.59	0.63
	(0.43–4.28)		(0.25–9.28)	
CVD	5.4	0.018	28.2	0.002
	(1.32–22.02)		(3.44–230.76)	
SOFA score	1.23	0.038	1.62	0.007
	(1.01–1.49)		(1.14–2.30)	
APACHE II score	1.07	0.221	1.04	0.663
	(0.95–1.20)		(0.86–1.26)	

A *p*-value < 0.05 indicates statistical significance. Abbreviations: OR, odds ratio; CI, confidence interval; CVD, cerebrovascular disease; SOFA, Sequential Organ Failure Assessment; APACHE, Acute Physiology and Chronic Health Evaluation II.

## Data Availability

The study dataset is available from the corresponding author upon reasonable request.

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
