# Peer review of "Colistin Monotherapy versus Colistin plus Sitafloxacin for Therapy of Carbapenem-Resistant Acinetobacter baumannii Infections: A Preliminary Study"

_antibiotics, 2022, doi:10.3390/antibiotics11121707_

Round 1

Reviewer 1 Report (Previous Reviewer 1)

The manuscript was clearly amended. All suggestions and recommendations were well-reflected.

Author Response

Thank you so much for your kind comments and suggestions.

Reviewer 2 Report (Previous Reviewer 3)

Dear authors,

All comments, observations and recommendations, as well as suggestions to improve the manuscript and have the quality to be published were resolved and corrected. As far as I am concerned, it satisfies my requirements and I consider that it can follow its process according to the indications of the journal editorial committee.

Congratulations.

Author Response

Thank you so much for your kind comments and suggestions.

Reviewer 3 Report (Previous Reviewer 4)

The number of patients included in the study is still small. In this situation, the conclusions cannot be supported by the results, from a statistical point of view. I recommend increasing the number of participants above the calculated minimum value.

Author Response

Thank you so much for your comments. This manuscript is only a preliminary study as said in the title of the manuscript. We also mentioned small sample size as one of the limitations of this study in the discussion section. Moreover, all outcomes of treament between both groups were very similar among 60% of the subjects according to the calculated sample size. Therefore, it is very unlikely to demonstrate significant differences in all outcomes if the study is carried on until the estimated sample size is met.

Round 2

Reviewer 3 Report (Previous Reviewer 4)

The main problem related to the study is that the conclusions are not supported by the results because the group is too small.

This manuscript is a resubmission of an earlier submission. The following is a list of the peer review reports and author responses from that submission.

Round 1

Reviewer 1 Report

I read, Colistin monotherapy versus colistin plus sitafloxacin for therapy of carbapenem-resistant Acinetobacter baumannii infections: A preliminary study with interest. In this manuscript, the authors aimed to compare efficacy and safety between colistin plus sitafloxacin with colistin alone for therapy for CR A. baumannii infection.

 I have some questions and suggestions.

1. Can you explain why this review is new or telling new things?

2. Why did the authors choose sitafloxacin in combination with colistin and why not choose another antibiotic?

3. What is the method for testing colistin susceptibility in your laboratory? However, in July 2016 EUCAST issued the warning, that broth microdilution (BMD) is the only valid method for the AST of colistin (https://www.eucast.org/fileadmin/src/media/PDFs/EUCAST_files/Warnings/Warnings_docs/Warning_-_colistin_AST.pdf ). The problem of the clinical management of patients affected by CRAB strains poses a number of questions about the correct execution and interpretation of colistin susceptibility tests, especially agar diffusion tests (e.g., E-test®, disk-diffusion) and some automated systems. Inappropriate colistin susceptibility testing can lead to misinterpretation of the results and, consequently, to an inadequate antibiotic therapy. Therefore, it is possible that infections with isolates classified as false-susceptible for colistin, were treated with colistin. This could be a significant factor contributing to the unfavorable outcome of patients treated with colistin. Please discussion and add antibiotics susceptibility test section.

4. Why did you decide to use 200 mg of sitafloxacin per day? Why didn't you select a dose that was higher or lower? However, the dosage of sitafloxacin varied and ranged from 100 mg per day to 400 mg per day.

5. Line 138: Please provide a detailed explanation of the colistin maintenance dose.

6. Line 140-141: Please provide a detailed description of colistin and sitafloxacin adjusted according to the individual patient's renal function.

7. Discussion is rather weak. The data from other studies is relatively small. Please add more data about colistin in combination with other drugs for the treatment of CRAB infection from the other studies as well for comparison with this study.

8. Could you please insert/use/discuss the issue of combination treatment of CRAB infection? Please see,

1. https://doi.org/10.3390/jcm11113239

2. https://doi.org/10.3390/pharmaceutics13020162

3. https://doi.org/10.3390/pharmaceutics14061266

9. Do you think that using inappropriate sitafloxacin doses resulted in the outcomes of studies showing no difference between monotherapy and combination therapy? And why do these study's findings differ from those of in vitro studies?

10. Have you tested sitafloxacin's MIC value? due to the possibility of a synergistic effect with combination drugs.

11. Please add more limitations to your study. For example, imbalance in some patients’ characteristics.

12. Please provide more data on the importance of physicians, pharmacists, and healthcare professionals around the world recognizing the efficacy and safety of colistin plus sitafloxacin.

Reviewer 2 Report

This study focused on the effect of colistin combine therapy. As many previous reports showed negative results, this study found no significant difference between the colistin monotherapy and the colistin plus sitafloxacin. Because of the importance of colistin treatment, the negative results still valuable.

However, as the authors’ mentioned, the dose and route of administration of sitafloxacin were both inappropriate. It’s a major flaw of the research design and make the results valueless for reference.

Reviewer 3 Report

Dear authors,

The main problem related with yor paper is the calculation of the sample size (47 individuals) and only included 28 patients, which makes it difficult to find differences according to the statistics proposed in your methodology.

Another design problem is the selection and exclusion criteria since they are not clear. They do not detail the record of the use of other antibiotics before patient recruitment and do not describe how the groups were assigned.

They do not describe the characteristics of the intervention (route of administration among other details), although they mention some aspects in the discussion.

I think it requires expanding the size of the sample and meeting the criteria for calculating the minimum size of the universe of study.

To consider publishing, working on some adjustments in the format and generation of charts and figures that will help a more visual interpretation of the results.  My specific recommendations have been added to the attached .pdf file for your consideration.

Although the research has biases related to your experimental design (is not blinded to eliminate selection bias), among other methodological aspects, I understand are related to the small universe of study and the clinical characteristics of this particular type of severe infectious disease with rapid evolution. Consider that as a preliminary report you need to suggest and specify in the text the next step of your research team, where it is recommended to design a more rigorously clinical trial with a larger universe of study and control more confusing variables.

On the other hand, although the information on the subject developed is scarce in the literature, it is recommended to carry out an extensive search in different collections and databases to expand the references since in a quick search I was able to find some authors with evidence and background. important for the theoretical framework.

Author Response

Reviewer 3

Comments and Suggestions for Authors

Dear authors,

The main problem related with your paper is the calculation of the sample size (47 individuals) and only included 28 patients, which makes it difficult to find differences according to the statistics proposed in your methodology.

Response: Thank you for your kind comments. Our sample size calculation assumed a 50% reduction in mortality in the combination group, which was likely too high. We decided to stop this study after enrollment of 60% of the patients because the mortality rates of the patients in both groups were identical and it was extremely unlikely to observe a 50% reduction in the mortality rate of the patients in the combination therapy group. These sentences are stated in the discussion.

Another design problem is the selection and exclusion criteria since they are not clear. They do not detail the record of the use of other antibiotics before patient recruitment and do not describe how the groups were assigned.

Response: Thank you for your useful comments. More exclusion criteria is added in the method section. The data of the concurrent antibiotics is stated in the Table 1. However, the prior use of other antibiotics is not collected. Investigator generated pre-determined table by block of 4 for group randomization.

They do not describe the characteristics of the intervention (route of administration among other details), although they mention some aspects in the discussion.

Response: Thank you for your comments, the dose and route of study drugs are stated in the method section.

I think it requires expanding the size of the sample and meeting the criteria for calculating the minimum size of the universe of study.

Response: Thank you for your kind comments. Our sample size calculation assumed a 50% reduction in mortality in the combination group, which was likely too high. We decided to stop this study after enrollment of 60% of the patients because the mortality rates of the patients in both groups were identical and it was extremely unlikely to observe a 50% reduction in the mortality rate of the patients in the combination therapy group. These sentences are stated in the discussion.

To consider publishing, working on some adjustments in the format and generation of charts and figures that will help a more visual interpretation of the results.  My specific recommendations have been added to the attached .pdf file for your consideration.

Response: Thank you for your kind supports. The manuscript is revised as your specific suggestions (please see the attached files “Specific response to reviewer 3”).

Although the research has biases related to your experimental design (is not blinded to eliminate selection bias), among other methodological aspects, I understand are related to the small universe of study and the clinical characteristics of this particular type of severe infectious disease with rapid evolution. Consider that as a preliminary report you need to suggest and specify in the text the next step of your research team, where it is recommended to design a more rigorously clinical trial with a larger universe of study and control more confusing variables.

Response: Thank you so much for your suggestions. The next experimental direction of study is stated in discussion section of manuscript as suggested. Further study comparing these two drug regimens, except using parenteral sitafloxacin instead of oral sitafloxacin, in a sufficiently large study population of patients with CR A. baumannii infection is needed.

Besides, please see the attachment.

Reviewer 4 Report

The manuscript entitled "Colistin monotherapy versus colistin plus sitafloxacin for therapy of carbapenem-resistant Acinetobacter baumannii infections: A preliminary study", is original and current, but the study group is too small. To improve the study, I recommend increasing the number of patients.
